# A sex-specific Mendelian randomization-phenome-wide association study of body mass index

**Zhu Liduzi Jiesisibieke[1], Io Ieong Chan[2], Jack Chun Man Ng[1], C Mary Schooling[1,3]***

[1]School of Public Health, Li Ka Shing Faculty of Medicine, The University of Hong Kong, Hong Kong, China; [2]Department of Public Health and Medicinal Administration, Faculty of Health Sciences, University of Macau, Macao, China; [3]Graduate School of Public Health and Health Policy, City University of New York, New York, United States

## eLife Assessment

The study presents **valuable** findings on Mendelian randomization-phenome-wide association, with BMI associated with health outcomes, and there is a focus on sex differences. The phenotype and genotype data are **convincing**. The work will be of interest to researchers and clinicians in epidemiology, public health and medicine.

**\*For correspondence:**
mary.schooling@sph.cuny.edu

**Competing interest:** The authors declare that no competing interests exist.

## Abstract

**Background:** Trials of incretins are making it increasingly clear that body mass index (BMI) is linked to several diseases throughout life, but trials cannot easily provide a comprehensive assessment of the role of BMI in health-related attributes for men and women. To systematically investigate the role of BMI, we conducted a sex-specific Mendelian randomization-phenome-wide association study.
**Methods:** We comprehensively examined the associations of genetically predicted BMI in women (*n*: 194,174) and men (*n*: 167,020) using health-related attributes from the UK Biobank with inverse variance weighting and sensitivity analysis.
**Results:** BMI impacted 232 of 776 traits considered in women and 203 of 680 traits in men, after adjusting for false discovery; differences by sex were found for 105 traits, and 46 traits remained after adjusting for false discovery. BMI was more strongly positively associated with myocardial infarction, major coronary heart disease events, ischemic heart disease, and heart attack in men than women. BMI was more strongly positively associated with apolipoprotein B (ApoB) and diastolic blood pressure in women than men.
**Conclusions:** Our study revealed that BMI might affect a wide range of health-related attributes and also highlights notable sex differences in its impact, including opposite associations for certain attributes, such as ApoB; and stronger effects in men, such as for cardiovascular diseases. Our findings underscore the need for nuanced, sex-specific policy related to BMI to address inequities in health.
**Funding:** None.

## Introduction

Global obesity prevalence more than doubled from 1980 to 2008 (*Finucane et al., 2011*). In 2022, one in eight people were obese worldwide (*World Health Organisation, 2024*). Body mass index (BMI) is a longstanding measure of obesity, despite its high specificity but low sensitivity (*Chooi et al., 2019*;

*Okorodudu et al., 2010*). Many observational studies have found BMI associated with disease risk factors and lifespan (*Whitlock et al., 2009*; *Kahn et al., 2006*; *Khan et al., 2018*). Generally, a higher BMI is detrimental to health. However, BMI does not account well for fat distribution. For men and women with the same BMI, women tend to store more fat (*Power and Schulkin, 2008*). Furthermore, men tend to store fat around their organs, while women are more likely to store it subcutaneously (*Power and Schulkin, 2008*), potentially leading to different health impacts by sex (*Costanzo et al., 2022*; *Power and Schulkin, 2008*). As such, the effect of BMI on health may vary between men and women and may be a modifiable factor contributing to differences in lifespan by sex.

Few previous observational studies have systematically assessed the role of BMI in health and disease, and are limited by their susceptibility to confounding (*Fewell et al., 2007*). Mendelian randomization (MR) studies reduce confounding by using genetic proxies, such as single-nucleotide polymorphisms (SNPs), for exposures (*Smith and Ebrahim, 2003*). Previous phenome-wide association studies using MR (MR-PheWASs) have identified impacts of sex-combined BMI on endocrine disorders, circulatory diseases, inflammatory and dermatological conditions, some biomarkers, and feelings of nervousness (*Hyppönen et al., 2019*; *Millard et al., 2019*; *Millard et al., 2015*), but did not systematically use sex-specific BMI for the exposure or sex-specific outcomes. Previous MR studies and trials of incretins have expanded our knowledge about a broad range of effects of BMI (*Larsson et al., 2020*; *Marso et al., 2016*). To our knowledge, no sex-specific PheWAS has investigated the effects of BMI on health outcomes (*Hyppönen et al., 2019*; *Millard et al., 2019*; *Millard et al., 2015*). To address this gap, we conducted a sex-specific PheWAS, using the largest available sex-specific GWAS of BMI, to explore the impact of sex-specific BMI on sex-specific health-related attributes.

## Methods
### Data sources
### Exposure: body mass index
To obtain genetic information for BMI sex-specifically, we used BMI (inverse rank normalized) from two sources: a prospective cohort study of half a million adults from the UK Biobank, and the Genetic Investigation of ANthropometric Traits (GIANT) consortium (*Locke et al., 2015*), a GWAS meta-analysis of 339,224 participants mostly of European ancestry. The UK Biobank has the advantage of a larger sample size and denser genotyping but is the only large-scale sex-specific source for many outcomes. The sex-specific UK Biobank BMI GWAS conducted by Neale Lab (women: 194,174; men: 167,020) was adjusted for age, age squared, and the first 20 principal components (https://www.nealelab.is/uk-biobank/faq). The GIANT GWAS has the advantage of using a different sample from the UK Biobank but is smaller with less dense genotyping (women: 171,977; men: 152,893). The sex-specific GIANT GWAS was adjusted for age, age squared, and study-specific covariates (*Locke et al., 2015*). We also considered overall BMI from GIANT (*n*: 681,275) which includes the UK Biobank participants (approximately 64%), and was adjusted for age, sex, and study-specific covariates (*Yengo et al., 2018*).

### Outcomes
The UK Biobank is currently the largest and most comprehensive source for sex-specific GWAS, provided by Neale Lab (women: 194,174; men: 167,020), including many disease outcomes and physiological attributes. The average age at recruitment of UK Biobank participants was 57 years, as described previously (*Collins, 2012*).

### Outcomes: inclusion and exclusion criteria
For continuous outcomes only rank normalized health attributes with at least 1000 participants (*Verma et al., 2018*) were included to maintain statistical power. For binary attributes only those with at least 200 cases were included, while duplicate phenotypes were excluded. Where GWAS of very similar attributes were provided, we only included one instance. Where an attribute has a standard measure, such as forced expiratory volume in 1 s, we used that in preference to similar measures. We also excluded attributes only assessed in selected subgroups of the UK Biobank participants, such as electrocardiogram, which was only conducted in healthy people (*Ramírez et al., 2021*). Where attributes were available from both self-report and doctor diagnosis, we used self-reports. This is because comprehensive record linkage to doctor diagnoses has not yet been fully implemented

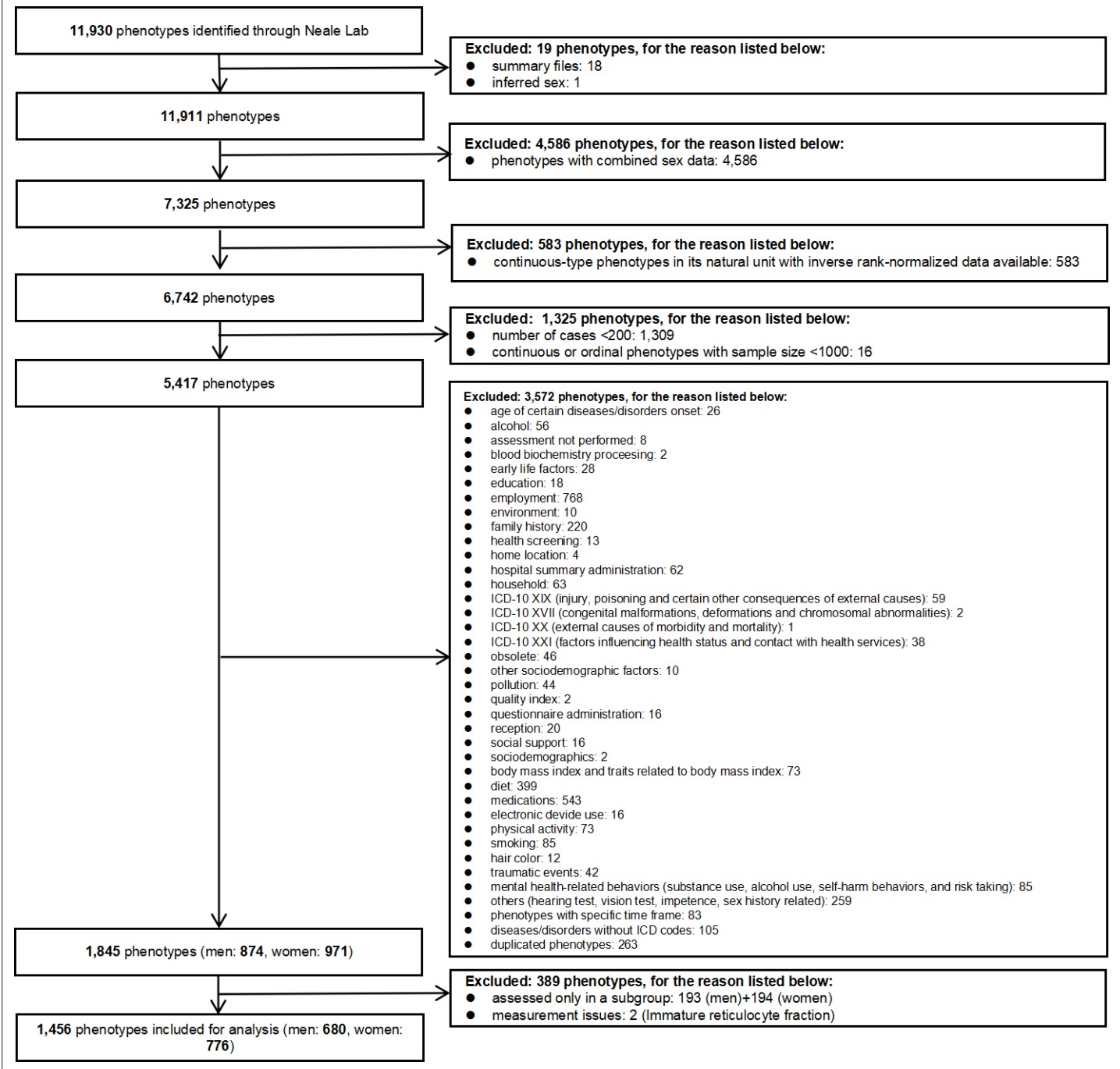

**Figure 1.** Selection criteria for the phenotypes.

for the UK Biobank, so information from doctor diagnoses may not fully represent the broader UK Biobank cohort. Finally, attributes with known measurement issues, such as estrogen and immature reticulocyte fraction, were excluded (*Newman and Handelsman, 2014*; *Piva et al., 2015*). Detailed exclusion criteria are listed in *Figure 1*.

## Outcomes: attribute categorization

We categorized attributes as age at recruitment, physical measures, lifestyle and environmental, medical conditions, operations, physiological factors, cognitive function, health and medical history, sex-specific factors, blood assays, and urine assays based on the UK Biobank categories (https://

biobank.ndph.ox.ac.uk/ukb/cats.cgi). Binary attributes with an International Classification of Diseases-10 code were categorized by International Classification of Diseases-10 chapter, as (I) infectious diseases, (II) neoplasms, (III) diseases of the hematopoietic system and blood disorders, (IV) endocrine and metabolic diseases, (V) psychological disorders, (VI) diseases of the nervous system, (VII and VIII) diseases of the sensory system (eyes and ears), (IX) diseases of the circulatory system, (X) diseases of the respiratory system, (XI) diseases of the digestive system, (XII) diseases of skin and subcutaneous tissue, (XIII) diseases of the musculoskeletal system and connective tissue, (XIV) diseases of the genitourinary system, (XV) pregnancy, childbirth, and the puerperium and (XVIII) symptoms.

## Statistical analysis

MR was used to assess effects of genetically predicted sex-specific BMI on each attribute considered sex-specifically. Inverse variance weighted estimates were used initially, that is, meta-analysis of Wald estimates (SNP on outcome divided by SNP on exposure), and then sensitivity analysis (MR-Egger) was used for any associations found. The significance level was adjusted for the false discovery to account for multiple comparisons (*Benjamini and Hochberg, 1995*). Specifically, we ranked the p-values for men and women, respectively, calculated the Benjamini–Hochberg (BH) value, and identified the significant attributes influenced by BMI using the BH value. We obtained differences by sex using a z-test (*Paternoster et al., 1998*), which as recommended was on a linear scale for dichotomous outcomes (*Knol et al., 2007*; *Rothman, 2008*), then we identified which ones remained after allowing

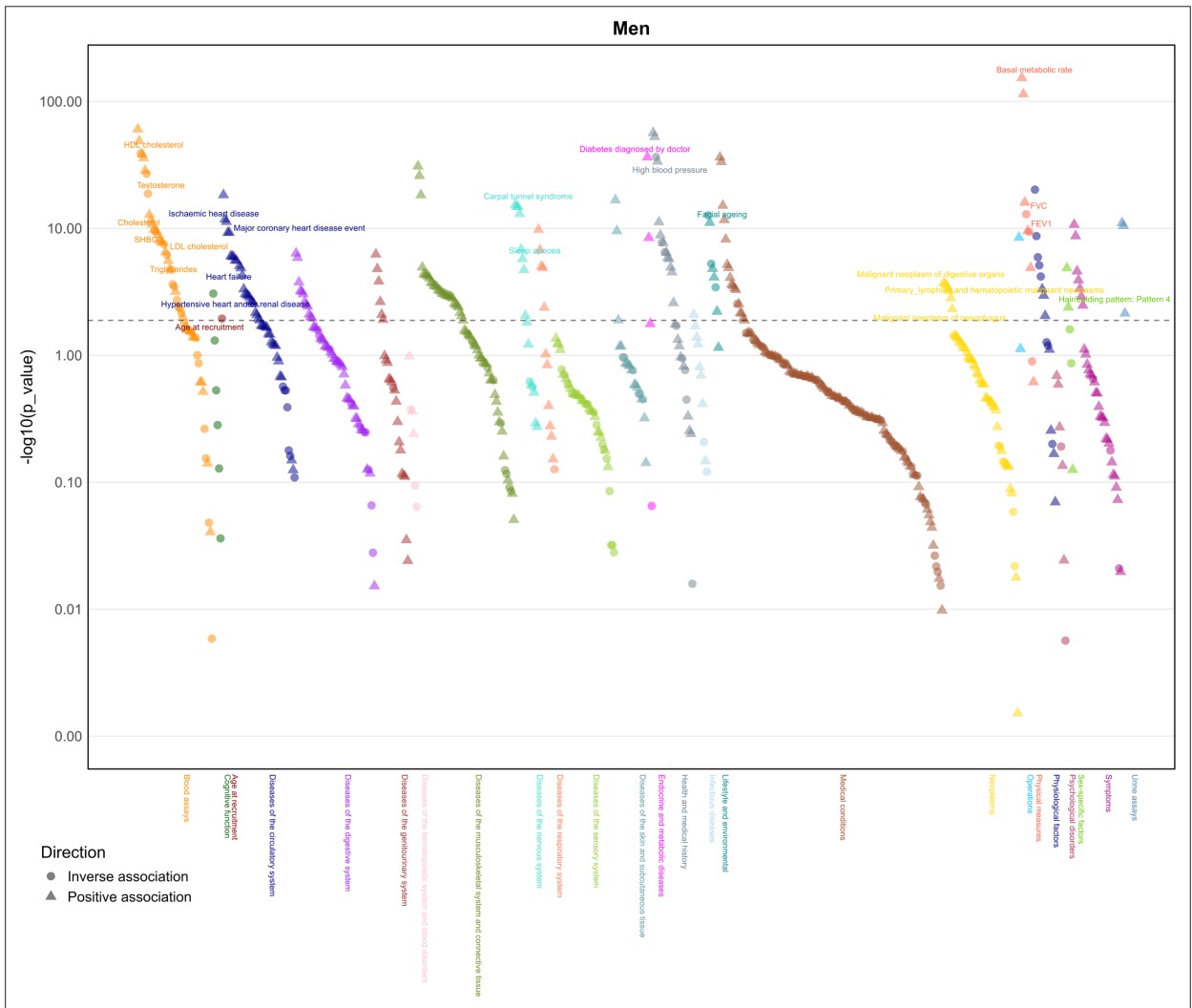

**Figure 2.** Manhattan plot of body mass index on phenotypes in men (*n*: 167,020) from the UK Biobank. Note: The dotted lines indicate the significant threshold.

for false discovery. We used the R packages 'TwoSampleMR' (version: 0.6.1), 'Mendelian Random-ization' (version: 0.10.0) for the MR analysis, and 'metafor' (version: 4.6-0) to assess sex differences.

## Results

Initial analysis using sex-specific BMI from GIANT yielded similar estimates as when using sex-specific BMI from the UK Biobank but had fewer SNPs resulting in wider confidence intervals (*Supplementary file 1a*) and fewer significant associations (*Supplementary file 1b*). Analysis using sex-combined GIANT yielded more significant associations but lacks granularity, so we presented the results obtained using sex-specific BMI from the UK Biobank.

### Sex-specific estimates

In men, BMI was associated with 203 of the 680 health-related attributes considered using false discovery (*Figure 2*, *Supplementary file 1b*). As expected, BMI was positively associated with ischemic heart disease, heart failure, hypertensive heart and/or renal disease, major coronary heart disease events, heart attack, diabetes, hypertension, sleep apnea, daytime dozing/sleeping (narcolepsy), triglycerides, urinary potassium, urinary sodium, and major surgeries. BMI was also inversely associated with age at recruitment, osteoporosis, hay fever and allergic rhinitis or eczema, high-density lipoprotein cholesterol (HDL-c), cholesterol, low-density lipoprotein cholesterol (LDL-c), sex hormone

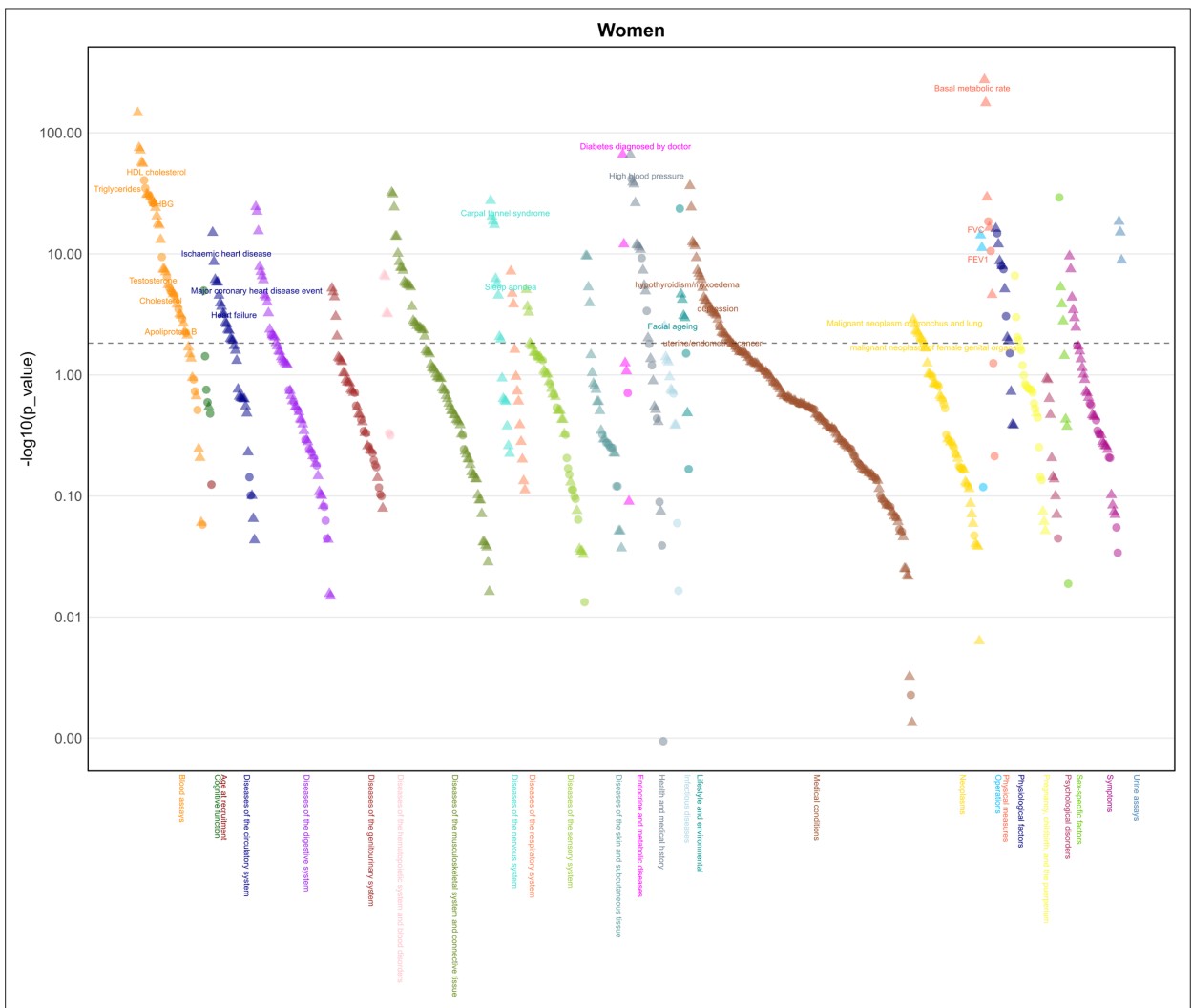

**Figure 3.** Manhattan plot of body mass index on phenotypes in women (*n*: 194,174) from the UK Biobank. Note: The dotted lines indicate the significant threshold.

binding globulin (SHBG), and total testosterone. Positive associations of BMI with higher risk of digestive system cancers, lymphomas, primary lymphoid and hematopoietic malignancies, and esophageal cancer were also found in men. Furthermore, we found higher BMI associated with accelerated facial aging and baldness.

In women, BMI was associated with 232 of 776 health-related attributes considered using false discovery (*Figure 3*, *Supplementary file 1b*). As expected, BMI was positively associated with ischemic heart disease, heart failure, hypertensive heart and/or renal disease, heart attack, diabetes, hypertension, sleep apnea, daytime dozing/sleeping (narcolepsy), apolipoprotein B (ApoB), triglycerides, total testosterone and urinary potassium and sodium, and major surgeries. BMI was also inversely associated with osteoporosis, HDL-c, cholesterol, LDL-c, and SHBG. BMI was positively associated with risk of uterine/endometria cancer, cancer of the bronchus and lung and intrathoracic organs, as well as cancers affecting the skin and female reproductive organs. Higher BMI in women was also associated with accelerated facial aging.

## Differences by sex

We found significant differences by sex in the associations of BMI with 105 health-related attributes (p-value <0.05); 46 phenotypes remained after allowing for false discovery (*Table 1*). Of these 46 differences most (35) were in magnitude but not direction, such as for SHBG, ischemic heart disease, heart attack, and facial aging, while 11 were directionally different.

Notably, BMI was more strongly positively associated with myocardial infarction, major coronary heart disease events, ischemic heart disease, heart attack, and facial aging in men than in women. BMI was more strongly positively associated with diastolic blood pressure, and hypothyroidism/myxoedema in women than men. BMI was more strongly inversely associated with LDL-c, hay fever, and allergic rhinitis or eczema in men than women. BMI was more strongly inversely associated with SHBG in women than men.

BMI was inversely associated with ApoB, iron deficiency anemia, hernia, and total testosterone in men, while positively associated with these traits in women (*Table 1*). BMI was inversely associated with sensitivity/hurt feelings, and ever seeking medical advice for nerves, anxiety, tension, or depression in men. However, BMI was positively associated with sensitivity/hurt feelings and ever seeking medical advice for these same issues in women. BMI was positively associated with muscle or soft tissue injuries and hemorrhage from respiratory passages in men, while inversely associated with these traits in women.

Of the 42 significant sex-specific associations identified in both the UK Biobank and the sex-specific GIANT consortium for men, all were directionally consistent. Similarly, for women, all 45 such significant associations were directionally consistent.

## Discussion

Consistent with previous studies, BMI was positively associated with many health-related attributes, such as ischemic heart disease, heart failure, hypertensive heart and/or renal disease, diabetes, hypertension, and sleep apnea. Our study adds by showing sex differences in some traits related to psychological disorders as well as ApoB and showing stronger associations in men than women for common cardiovascular diseases.

### Comparison with previous studies

Consistent with previous MR studies, BMI was positively associated with the risk of ischemic heart disease, heart failure, hypertensive heart disease (*Riaz et al., 2018*), hypertension (*Riaz et al., 2018*), and diabetes (*Larsson and Burgess, 2021*) in both men and women as would be expected. A previous MR study found height and fat-free mass positively associated with follicular lymphoma (*Zhou et al., 2024*). However, we found BMI positively associated with lymphomas in men, but not in women (although directionally consistent and not significant after allowing for false discovery). Another MR study found BMI positively associated with asthma and poorer lung function, but not with hay fever (*Skaaby et al., 2018*), while we found BMI inversely associated with hay fever and allergy allergic rhinitis or eczema in men, but not in women. Previous studies have shown BMI inversely associated with sodium/potassium (*Zanetti et al., 2020*), while we found BMI positively associated with urinary

**Table 1.** Sex differences of body mass index on phenotypes in men (*n*: 167,020) and women (*n*: 194,174).

Red: Stronger associations in women and different directions in men and women; Green: Stronger associations in men and different directions in men and women; Orange: Stronger associations in women and same directions in men and women; Blue: Stronger associations in men and same directions in men and women.

| Category | Phenotypes | Men | Women | Sex difference after false-discovery rate |
|---|---|---|---|---|
| Blood assays | SHBG | ↓ | ↓ | √ |
| | Albumin | ↓ | ↓ | × |
| | Vitamin D | ↓ | ↓ | √ |
| | Cholesterol | ↓ | ↓ | × |
| | Total bilirubin | ↓ | ↓ | × |
| | LDL cholesterol | ↓ | ↓ | √ |
| | Direct bilirubin | ↓ | ↓ | √ |
| | Platelet count | ↓ | ↑ | × |
| | Testosterone | ↓ | ↑ | √ |
| | Apoliprotein B | ↓ | ↑ | √ |
| | Platelet crit | ↓ | ↑ | √ |
| | Aspartate aminotransferase | ↑ | ↑ | × |
| | Alkaline phosphatase | ↑ | ↑ | × |
| | Neutrophill count | ↑ | ↑ | × |
| | Red blood cell (erythrocyte) distribution width | ↑ | ↑ | × |
| | Triglycerides | ↑ | ↑ | × |
| | Urate | ↑ | ↑ | √ |
| | C-reactive protein | ↑ | ↑ | √ |
| Diseases of the circulatory system | Diagnoses – main ICD10: I35 Nonrheumatic aortic valve disorders | ↑ | ↑ | × |
| | Peripheral artery disease | ↑ | ↑ | √ |
| | Death due to cardiac causes | ↑ | ↑ | × |
| | Heart failure, strict | ↑ | ↑ | × |
| | Diagnoses – main ICD10: I26 Pulmonary embolism | ↑ | ↑ | × |
| | DVT of lower extremities and pulmonary embolism | ↑ | ↑ | × |
| | Venous thromboembolism | ↑ | ↑ | × |
| | Myocardial infarction, strict | ↑ | ↑ | √ |
| | Major coronary heart disease event | ↑ | ↑ | √ |
| | Major coronary heart disease event excluding revascularizations | ↑ | ↑ | √ |
| | Coronary atherosclerosis | ↑ | ↑ | √ |
| | Ischemic heart disease, wide definition | ↑ | ↑ | √ |
| | Diseases of the circulatory system | ↑ | ↑ | × |

*Table 1 continued on next page*

*Table 1 continued*

| Category | Phenotypes | Men | Women | Sex difference after false-discovery rate |
|---|---|---|---|---|
| Diseases of the digestive system | Diagnoses – main ICD10: K60 Fissure and fistula of anal and rectal regions | ↑ | ↑ | × |
| | Diagnoses – main ICD10: K81 Cholecystitis | ↑ | ↑ | × |
| | Hernia | ↓ | ↑ | √ |
| | Diagnoses – main ICD10: K80 Cholelithiasis | ↑ | ↑ | √ |
| | Disorders of gallbladder, biliary tract, and pancreas | ↑ | ↑ | √ |
| | Diseases of the digestive system | ↑ | ↑ | √ |
| Diseases of the genitourinary system | Diagnoses – main ICD10: N17 Acute renal failure | ↑ | ↑ | × |
| | Diagnoses – main ICD10: N39 Other disorders of urinary system | ↑ | ↑ | √ |
| Diseases of the hematopoietic system and blood disorders | Other anemias | ↓ | ↑ | × |
| | Other and unspecified anemias | ↓ | ↑ | × |
| | Iron deficiency anemia | ↓ | ↑ | √ |
| | Diseases of the blood and blood-forming organs and certain disorders involving the immune mechanism | ↑ | ↑ | √ |
| Diseases of the musculoskeletal system and connective tissue | Cervicalgia | ↓ | ↑ | × |
| | Soft tissue disorders related to use, overuse, and pressure | ↑ | ↑ | × |
| | Impingement syndrome of shoulder | ↑ | ↑ | × |
| | Pain in limb | ↑ | ↑ | √ |
| | Other soft tissue disorders, not elsewhere classified | ↑ | ↑ | √ |
| | Diagnoses – main ICD10: M79 Other soft tissue disorders, not elsewhere classified | ↑ | ↑ | √ |
| Diseases of the nervous system | Diagnoses – main ICD10: G57 Mononeuropathies of lower limb | ↓ | ↑ | × |
| | Sleep apnea | ↑ | ↑ | × |
| | Diagnoses – main ICD10: G47 Sleep disorders | ↑ | ↑ | × |
| | Carpal tunnel syndrome | ↑ | ↑ | × |
| | Nerve, nerve root, and plexus disorders | ↑ | ↑ | × |
| Diseases of the respiratory system | Diagnoses – main ICD10: J18 Pneumonia, organism unspecified | ↑ | ↑ | √ |
| | Diseases of the respiratory system | ↑ | ↑ | × |
| Diseases of the sensory system | Diagnoses – main ICD10: H61 Other disorders of external ear | ↑ | ↓ | × |
| | Degeneration of macula and posterior pole | ↓ | ↑ | × |
| Diseases of the skin and subcutaneous tissue | Diagnoses – main ICD10: L03 Cellulitis | ↑ | ↑ | √ |
| Endocrine and metabolic diseases | Diabetes diagnosed by doctor | ↑ | ↑ | √ |
| Health and medical history | Blood clot, DVT, bronchitis, emphysema, asthma, rhinitis, eczema, allergy diagnosed by doctor: hay fever, allergic rhinitis, or eczema | ↓ | ↓ | √ |
| | Blood clot, DVT, bronchitis, emphysema, asthma, rhinitis, eczema, allergy diagnosed by doctor: none of the above | ↑ | ↓ | √ |
| | Vascular/heart problems diagnosed by doctor: heart attack | ↑ | ↑ | √ |
| | Chest pain or discomfort | ↑ | ↑ | × |
| | Mouth/teeth dental problems: dentures | ↑ | ↑ | × |

*Table 1 continued on next page*

*Table 1 continued*

| Category | Phenotypes | Men | Women | Sex difference after false-discovery rate |
|---|---|---|---|---|
| Lifestyle and environmental | Morning/evening person (chronotype) | ↓ | ↓ | × |
| | Facial aging | ↑ | ↑ | √ |
| Medical conditions | Non-cancer illness code, self-reported: osteoporosis | ↓ | ↓ | × |
| | Non-cancer illness code, self-reported: headaches (not migraine) | ↓ | ↑ | × |
| | Non-cancer illness code, self-reported: pernicious anemia | ↓ | ↑ | × |
| | Non-cancer illness code, self-reported: psoriatic arthropathy | ↓ | ↑ | × |
| | Non-cancer illness code, self-reported: eye/eyelid problem | ↓ | ↑ | × |
| | Non-cancer illness code, self-reported: allergy/hypersensitivity/anaphylaxis | ↓ | ↑ | × |
| | Non-cancer illness code, self-reported: ulcerative colitis | ↑ | ↓ | × |
| | Non-cancer illness code, self-reported: bladder problem (not cancer) | ↓ | ↑ | × |
| | Non-cancer illness code, self-reported: *Helicobacter pylori* | ↑ | ↑ | × |
| | Non-cancer illness code, self-reported: muscle or soft tissue injuries | ↑ | ↓ | √ |
| | Non-cancer illness code, self-reported: muscle/soft tissue problem | ↓ | ↑ | × |
| | Non-cancer illness code, self-reported: gout | ↑ | ↑ | √ |
| | Non-cancer illness code, self-reported: cervical spondylosis | ↑ | ↑ | × |
| | Non-cancer illness code, self-reported: arthritis (nos) | ↑ | ↑ | × |
| | Non-cancer illness code, self-reported: unclassifiable | ↓ | ↑ | × |
| | Non-cancer illness code, self-reported: cholelithiasis/gall stones | ↑ | ↑ | √ |
| | Non-cancer illness code, self-reported: depression | ↑ | ↑ | × |
| | Non-cancer illness code, self-reported: hypothyroidism/myxoedema | ↑ | ↑ | √ |
| | Non-cancer illness code, self-reported: gastro-esophageal reflux (gord) / gastric reflux | ↑ | ↑ | × |
| | Non-cancer illness code, self-reported: angina | ↑ | ↑ | √ |
| | Non-cancer illness code, self-reported: asthma | ↑ | ↑ | √ |
| | Non-cancer illness code, self-reported: osteoarthritis | ↑ | ↑ | × |
| Neoplasms | Malignant neoplasm of urinary organs | ↑ | ↓ | × |
| | Diagnoses – main ICD10: D35 Benign neoplasm of other and unspecified endocrine glands | ↑ | ↑ | × |
| | Malignant neoplasm of colon | ↑ | ↓ | × |
| | Lymphomas | ↑ | ↑ | × |
| | Malignant neoplasm of digestive organs | ↑ | ↑ | × |
| Physical measures | Hand grip strength (right) | ↑ | ↓ | × |
| | Diastolic blood pressure, automated reading | ↑ | ↑ | √ |
| Physiological factors | Sensitivity/hurt feelings | ↓ | ↑ | √ |
| | Neuroticism score | ↓ | ↑ | × |
| | Seen doctor (GP) for nerves, anxiety, tension, or depression | ↓ | ↑ | √ |
| | Miserableness | ↑ | ↑ | √ |
| | Loneliness, isolation | ↑ | ↑ | √ |
| | Fed-up feelings | ↑ | ↑ | √ |

*Table 1 continued*

| Category | Phenotypes | Men | Women | Sex difference after false-discovery rate |
|---|---|---|---|---|
| Symptoms | Diagnoses – main ICD10: R04 Hemorrhage from respiratory passages | ↑ | ↓ | √ |
| | Diagnoses – main ICD10: R31 Unspecified hematuria | ↑ | ↓ | √ |
| Urine assays | Potassium in urine | ↑ | ↑ | × |

potassium and sodium. Additionally, we found a positive association of BMI with major surgeries. Consistent with a previous MR study (*Ardissino et al., 2022*), we found BMI positively associated with sleep apnea. We also found positive associations of BMI with daytime dozing in both sexes, which may indicate effects of BMI on quality of life. Higher BMI was also associated with a younger age at recruitment among men, suggesting poorer survival to recruitment in men than women given the UK Biobank had a short recruitment window making period effects unlikely. The association of BMI with age at recruitment was less evident in women, with no significant sex difference. Consistently, a prior observational study found high BMI more detrimental to lifespan in men than women, especially at younger ages (*Fontaine et al., 2003*).

## Differences in the associations of BMI with health-related attributes in men and women

BMI was more strongly associated with common cardiovascular diseases in men than women. BMI was positively associated with ApoB in women, whereas in men, the association was inverse (but not significant after correction for multiple comparisons). However, the sex difference was significant. This pattern contrasts with earlier MR studies that reported an inverse association of BMI with ApoB in the general population (*Bell et al., 2022*). BMI being more strongly associated with heart disease in men than women while also being protective for ApoB, a key heart disease risk factor, requires some explanation. BMI is a complex phenotype that may represent different attributes or have different consequences in men compared with women. Fat mass storage tends to differ between men and women (visceral versus subcutaneous) (*Power and Schulkin, 2008*). The causes of higher BMI may also differ between men and women, due to occupational roles, gender-based food preferences or cultural norms (*Kanter and Caballero, 2012*). Alternatively, unknown factors affected by BMI could contribute to heart disease specifically to men. Whether the difference in ischemic heart disease rates between men and women that emerged in the US and the UK the late 19th century (*Nikiforov and Mamaev, 1998*) is explained by rising BMI remains to be determined. In terms of psychological disorders, women being more affected psychologically by higher BMI is consistent with sociocultural pressures and norms surrounding women's rather than men's bodies (*Esnaola et al., 2010*; *Schwartz and Brownell, 2004*). We also found that BMI was associated with balding in men but not women, although the sex difference was not significant. Moreover, BMI was positively associated with facial aging in both men and women, with larger effects in men.

## Strengths and limitations

A major strength of this study was the focus on sex differences, which have not been explicitly explored in previous PheWAS of BMI, despite differences in lifespan by sex. We also considered false discovery for sex differences. Despite the large sample size and the broad range of outcomes considered in this study, limitations and concerns exist. First, MR has stringent assumptions, that is, relevance, independence, and exclusion restriction. The $F$-statistics were above 10, which addresses relevance. MR is largely free from confounding by design. For a few attributes (37 out of 1456), the MR-Egger intercept was significant while the IVW estimate was not (*Supplementary file 1c*), potentially indicating horizontal pleiotropic effects or the play of chance, which requires further investigation. Second, MR assesses lifetime effects of BMI, which may differ from those of short-term weight changes or weight cycling. Third, not all attributes were included. For example, heart rate from electrocardiogram and age at asthma diagnosis were excluded, because such information was limited to selected subgroups. Fourth, given the study's exploratory nature, we allowed for multiple comparisons to minimize chance findings, which means some BMI effects may not be captured. However,

such comprehensive consideration may identify previously unknown or ignored impacts, such as facial aging and balding. Fifth, while the UK Biobank does not fully represent the UK population, representativeness is not crucial for causal inference as long as the sample is not selected on exposure and outcome (*Greenland, 2003*). Sixth, focusing on a European population may limit generalizability, but it helps reduce bias from population stratification. Seventh, we excluded replicated or very similar attributes, potentially missing some attributes. However, this reduces false negatives when adjusting for multiple comparisons. Eighth, this study is open to selection bias because genetic endowment is lifelong but the UK Biobank participants are middle-aged or older, so potential recruits who did not live to recruitment because of their BMI are missing. This may bias estimates toward the null or inverse for harmful binary outcomes (*Thompson et al., 2013*), but is less likely to affect continuous outcomes (*Smit et al., 2019*). Ninth, we focused on BMI, because it is well accepted and easy to measure, although waist-to-hip ratio may be a better marker for mortality (*Khan et al., 2023*; *Ruder, 2023*). Tenth, although this study primarily utilized sex-specific BMI, we also conducted analyses using overall BMI from GIANT including the UK Biobank, which gave a generally similar interpretation (*Supplementary file 1a*). Using sex-specific BMI from the UK Biobank and GIANT may lead to lower statistical power than using overall population BMI but allows for the detection of traits that are affected differently by BMI by sex. Including findings from the overall population BMI from sex-combined GIANT (*Supplementary file 1a*) makes the results more comparable to previous similar studies. Eleventh, our study did not find associations of BMI with some early-onset cancers possibly because early-onset cancers tend to be of germline origin (*Qing et al., 2020*). Twelfth, we used information from Neale Lab, whose quality check removed participants whose self-reported sex differed from their biological sex, so our findings only relate to *cis* men and women. Thirteenth, while overlapping samples for BMI and the outcomes pose a potential issue in the main analysis, we also replicated the analysis using sex-specific instruments for BMI independent of the UK Biobank. Results were similar, suggesting bias from overlapping samples is minimal. Consistently, overlapping samples have been shown to make most difference for relatively small samples and for MR-Egger estimates, particularly when the $I^2_{GX}$ is low, leading to confounded estimates (*Minelli et al., 2021*). Lastly, we used linear MR, so we cannot exclude the possibility of a 'J' shaped association, however trials of incretins suggest a 'J' curve may be a manifestation of bias (*Wilding et al., 2021*).

## Public health implications

Trials of incretins have clearly indicated many benefits of weight loss particularly for older people (*Leiter et al., 2019*; *Lincoff et al., 2023*). Our study showing how BMI might increase risk of chronic diseases, reduce quality of life, and depress mental health in women, underscores the importance of addressing the obesity epidemic equitably in men and women, given differences by sex in some consequences of overweight/obesity, such as psychological disorders and lipids. Being overweight or obese is more common in men than women in developed countries (*Kanter and Caballero, 2012*; *Maruyama and Nakamura, 2018*). With ongoing global economic development, the same pattern is likely to become more common. Currently, women are more likely to seek medical intervention for weight management, such as semaglutide, than men (*Luthra, 2023*). To promote population health, it is important to address the unique challenges faced by overweight men and women. Men are at higher risk of heart disease and premature death due to high BMI, so targeted public health interventions, such as sex-specific weight recommendations or guidelines, could perhaps be considered.

## Conclusion

Our contemporary systematic examination found BMI associated with a broad range of health-related attributes. We also found significant sex differences in many traits, such as for cardiovascular diseases, underscoring the importance of addressing higher BMI in both men and women possibly as means of redressing differences in life expectancy. Ultimately, our study emphasizes the harmful effects of obesity and the importance of nuanced, sex-specific policy related to BMI to address inequities in health.

## Acknowledgements

We would like to thank the UK Biobank (Neale Lab) and Genetic Investigation of ANthropometric Traits (GIANT) for providing GWAS summary-level statistics. This research received no specific grant from any funding agency in the public, commercial, or not-for-profit sectors.

## Additional information

### Funding
No external funding was received for this work.

### Author contributions
Zhu Liduzi Jiesisibieke, Conceptualization, Data curation, Software, Formal analysis, Visualization, Methodology, Writing – original draft, Writing – review and editing; Io Ieong Chan, Jack Chun Man Ng, Data curation, Methodology, Writing – review and editing; C Mary Schooling, Conceptualization, Resources, Data curation, Software, Formal analysis, Supervision, Visualization, Methodology, Writing – original draft, Project administration, Writing – review and editing

### Author ORCIDs
Zhu Liduzi Jiesisibieke ⓘD https://orcid.org/0000-0002-4986-653X
Io Ieong Chan ⓘD https://orcid.org/0000-0001-9619-6535
Jack Chun Man Ng ⓘD https://orcid.org/0000-0002-3084-236X
C Mary Schooling ⓘD https://orcid.org/0000-0001-9933-5887

### Ethics
The study protocol was not pre-registered. We used only publicly available summary-level data and did not collect any original data in this study. Ethics approval and consent from individual participants can be found in the original publications.

Reviewer #2 (Public review): https://doi.org/10.7554/eLife.102573.3.sa1
Author response https://doi.org/10.7554/eLife.102573.3.sa2

## Additional files

### Supplementary files
Supplementary file 1. Sex-specific UK Biobank phewas estimates (a), counts of significant associations (b), and MR-Egger estimates (c). (a) Phenome-wide association analysis of causal estimates for body mass index from the UK Biobank (women: 194,174; men: 167,020) and sex-specific and sex-combined Genetic Investigation of ANthropometric Traits (GIANT) (sex-specific: women: 171,977, men: 152,893; sex-combined: 681,275) on attributes (women: 194,174; men: 167,020) from the UK Biobank based on inverse variance weighting. (b) Number of significant associations of body mass index from UK Biobank (women: 194,174; men: 167,020) and sex-specific and sex-combined GIANT (sex-specific: women: 171,977, men: 152,893; sex-combined: 681,275) with phenotypes from UK Biobank (women: 194,174; men: 167,020) by category. (c) MR-Egger sensitivity analysis of phenome-wide association causal estimates for body mass index on various phenotypes in women ($n$: 194,174) and men ($n$: 167,020) from the UK Biobank. Abbreviations used in supplementary files: lrnt: inverse rank normalized; SHBG: sex hormone binding globulin; GIANT: Genetic Investigation of ANthropometric Traits; GV: genetic variant; GWAS: genome-wide association study; ICD: International Classification of Diseases.

MDAR checklist

### Data availability
This study used data from the MR-base platform (https://www.mrbase.org/), the UK Biobank (http://www.nealelab.is/uk-biobank/) and the Genetic Investigation of ANthropometric Traits (GIANT) consortium.

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
