## [Editor Report · eLife Assessment]

The study presents **valuable** findings on Mendelian randomization-phenome-wide association, with BMI associated with health outcomes, and there is a focus on sex differences. The phenotype and genotype data are **convincing**. The work will be of interest to researchers and clinicians in epidemiology, public health and medicine.

---

## [Referee Report · Reviewer #2 (Public review)]

Summary:

In this present Mendelian randomization-phenome-wide association study, the authors found BMI to be positively associated with many health-related conditions, such as heart disease, heart failure, and hypertensive heart disease. They also found sex differences in some traits, such as cancer, psychological disorders, and ApoB.

Strengths:

The use of the UK-biobank study with detailed phenotype and genotype information.

Comments on revisions:

I believe the authors have presented convincing arguments for the novelty and interpretation of their study. I have no additional comments.

---

## [Author Response]

The following is the authors’ response to the original reviews

**eLife Assessment**
The study presents some useful findings on Mendelian randomization-phenome-wide association, with BMI associated with health outcomes, and there is a focus on sex differences. Although there are some solid phenotype and genotype data, some of the data are incomplete and could be better presented, perhaps benefiting from more rigorous approaches. Confirmation and further assessment of the observed sex differences will add further value.

Thank you for your positive comments. We have revised the analysis based on your feedback and that from the two reviewers. Specifically, we implemented a stricter multiple testing correction approach, improved the figures, included additional figures in the Supplementary Materials, considered the sex differences more rigorously and reported them in more detail. A comprehensive description of the revisions is provided below.

**Public Reviews:**

**Reviewer #1 (Public review):**
Summary:This study uses information from the UK Biobank and aims to investigate the role of BMI on various health outcomes, with a focus on differences by sex. They confirm the relevance of many of the well-known associations between BMI and health outcomes for males and females and suggest that associations for some endpoints may differ by sex. Overall their conclusions appear supported by the data. The significance of the observed sex variations will require confirmation and further assessment.Strengths:This is one of the first systematic evaluations of sex differences between BMI and health outcomes. The hypothesis that BMI may be associated with health differentially based on sex is relevant and even expected. As muscle is heavier than adipose tissue, and as men typically have more muscle than women, as a body composition measure BMI is sometimes prone to classifying even normal weight/muscular men as obese, while this measure is more lenient when used in women. Confirmation of the many well-known associations is as expected and attests to the validity of their approach. Demonstration of the possible sex differences is interesting, with this work raising the need for further study.

Thank you for your valuable comments. We are grateful for the time and effort you have devoted to reviewing our manuscript. We have strengthened our paper by adding your insightful comment about the rationale for sex-specific analysis to the introduction:

Weaknesses:(1) Many of the statistical decisions appeared to target power at the expense of quality/accuracy. For example, they chose to use self-reported information rather than doctor diagnoses for disease outcomes for which both types of data were available.

Thank you for your valuable comments. We apologize for the lack of clarity in our original description of the phenotypes. Information about health in the UK Biobank was obtained at baseline from tests, measurements and self reports. Subsequently comprehensive data linkage to hospital admissions, death registries and cancer registries was implemented. However, data linkage to primary care data, such as doctor diagnoses, has not been comprehensively implemented for the UK Biobank, possibly for logistic reasons. Doctor diagnoses are only available for about half the cohort, (https://www.ukbiobank.ac.uk/enable-your-research/about-our-data/health-related-outcomes-data). So, we used self-reported diagnoses because they are substantially more comprehensive than the doctor diagnoses. We have explained this point by making the following change to the Methods:

“Where attributes were available from both self-report and doctor diagnosis, we used self-reports. This is because comprehensive record linkage to doctor diagnoses has not yet been fully implemented for the UK Biobank, so information from doctor diagnoses may not fully represent the broader UK Biobank cohort.”

(2) Despite known problems and bias arising from the use of one sample approach, they chose to use instruments from the UK Biobank instead of those available from the independent GIANT GWAS, despite the difference in sample size being only marginally greater for UKB for the context. With the way the data is presented, it is difficult to assess the extent to which results are compatible across approaches.

Thank you for your comments. We agree completely about the issues with a one sample approach, please accept our apologies for not explaining our rationale. The sex-specific GIANT GWAS study is similar in size to the UK Biobank GWAS. However, the sex-specific GIANT GWAS is much less densely genotyped (~2,5 million variants) than the sex-specific UK Biobank GWAS (~10 million variants), so has less power, hence our use of the UK Biobank. To make this clear, we have added the number of variants in each study to the method section. Nevertheless, we also repeated analysis using sex-specific GIANT, as now given in the methods by making the following change

We amended the description in the first paragraph of the results section:

“Initial analysis using sex-specific BMI from GIANT yielded similar estimates as when using sex-specific BMI from the UK Biobank but had fewer SNPs resulting in wider confidence intervals (S Table 1) and fewer significant associations (S Table 1). Analysis using sex-combined GIANT yielded more significant associations but lacks granularity, so we presented the results obtained using sex-specific BMI from the UK Biobank.”

In the discussion we also made the following changes:

“Tenth, although this study primarily utilized sex-specific BMI, we also conducted analyses using overall BMI from GIANT including the UK Biobank, which gave a generally similar interpretation (S Table 1). Using sex-specific BMI from the UK Biobank and GIANT may lead to lower statistical power than using overall population BMI but allows for the detection of traits that are affected differently by BMI by sex. Including findings from the overall population BMI from sex-combined GIANT (S Table 1) makes the results more comparable to previous similar studies.”

(3) The approach to multiple testing correction appears very lenient, although the lack of accuracy in the reporting makes it difficult to know what was done exactly. The way it reads, FDR correction was done separately for men, and then for women (assuming that the duplication in tests following stratification does not affect the number of tests). In the second stage, they compared differences by sex using Z-test, apparently without accounting for multiple testing.

Thank you, we have accounted for multiple comparisons when considering differences by sex and have made corresponding changes. Specifically, in the methods, we changed:

“We obtained differences by sex using a z-test (Paternoster et al., 1998), which as recommended was on a linear scale for dichotomous outcomes (Knol et al., 2007; Rothman, 2008), then we identified which ones remained after allowing for false discovery”

We have made the following changes to the results section:

“We found significant differences by sex in the associations of BMI with 105 health-related attributes (p-value<0.05); 46 phenotypes remained after allowing for false discovery (Table 1). Of these 46 differences most (35) were in magnitude but not direction, such as for SHBG, ischemic heart disease, heart attack, and facial aging, while 11 were directionally different.

Notably, BMI was more strongly positively associated with myocardial infarction, major coronary heart disease events, ischemic heart disease, heart attack, and facial aging in men than in women. BMI was more strongly positively associated with diastolic blood pressure, and hypothyroidism/myxoedema in women than men. BMI was more strongly inversely associated with LDL-c, hay fever and allergic rhinitis in men than women. BMI was more strongly inversely associated with SHBG in women than men.

BMI was inversely associated with ApoB, iron deficiency anemia, hernia, and total testosterone in men, while positively associated with these traits in women (Table 1). BMI was inversely associated with sensitivity/hurt feelings, and ever seeking medical advice for nerves, anxiety, tension, or depression in men. However, BMI was positively associated with sensitivity/hurt feelings and ever seeking medical advice for these same issues in women. BMI was positively associated with muscle or soft tissue injuries and haemorrhage from respiratory passages in men, whilst inversely associated with these traits in women.”

We have correspondingly amended the discussion to reflect these changes by adding:

“Whether the difference in ischemic heart disease rates between men and women that emerged in the US and the UK the late 19th century (Nikiforov & Mamaev, 1998) is explained by rising BMI remains to be determined.”

(4) Presentation lacks accuracy in a few places, hence assessment of the accuracy of the statements made by the authors is difficult.

Thank you, we have revised the whole manuscript in order to improve clarity.

(5) Conclusion (Abstract) "These findings highlight the importance of retaining a healthy BMI" is rather uninformative, especially as they claim that for some attributes the effects of BMI may be opposite depending on sex/gender.

Thank you for your comments. We have changed the conclusion of the abstract, as given below:

“Our study revealed that BMI might affect a wide range of health-related attributes and also highlights notable sex differences in its impact, including opposite associations for certain attributes, such as ApoB; and stronger effects in men, such as for cardiovascular diseases. Our findings underscore the need for nuanced, sex-specific policy related to BMI to address inequities in health.”.

We have changed the Impact statement, as given below:

“BMI may affect a wide range of health-related attributes and there are notable sex differences in its impact, including opposite associations for certain attributes, such as ApoB; and stronger effects in men, such as for cardiovascular diseases. Our findings underscore the need for nuanced, sex-specific policy related to BMI.”

We have changed the conclusion of the paper, as given below:

“Our contemporary systematic examination found BMI associated with a broad range of health-related attributes. We also found significant sex differences in many traits, such as for cardiovascular diseases, underscoring the importance of addressing higher BMI in both men and women possibly as means of redressing differences in life expectancy. Ultimately, our study emphasizes the harmful effects of obesity and the importance of nuanced, sex-specific policy related to BMI to address inequities.in health.”

**Reviewer #2 (Public review):**
Summary:In this present Mendelian randomization-phenome-wide association study, the authors found BMI to be positively associated with many health-related conditions, such as heart disease, heart failure, and hypertensive heart disease. They also found sex differences in some traits such as cancer, psychological disorders, and ApoB.Strengths:The use of the UK-biobank study with detailed phenotype and genotype information.

Thank you for your valuable comments. We are grateful for the time and effort you have devoted to reviewing our manuscript.

Weaknesses:(1) Previous studies have performed this analysis using the same cohort, with in-depth analysis. See this paper: Searching for the causal effects of body mass index in over 300,000 participants in UK Biobank, using Mendelian randomization. https://journals.plos.org/plosgenetics/article?id=10.1371/journal.pgen.10079i51

Thank you for your valuable comments. We checked the paper carefully. It gives sex-specific estimates when the outcome was assessed in different ways in men and women, for example the question about number of children was asked in terms of live births in women and number of children fathered in men. In addition, for some significant findings, the authors investigated differences by sex. However, the paper did not use sex-specific BMI or sex-specific outcomes systematically. We have added this paper to our introduction and amended the text to explain the novelty of our study compared to previous studies.

“Previous phenome-wide association studies using MR (MR-PheWASs) have identified impacts of sex-combined BMI on endocrine disorders, circulatory diseases, inflammatory and dermatological conditions, some biomarkers and feelings of nervousness (Hyppönen et al., 2019; Millard et al., 2015; Millard et al. 2019), but did not systematically use sex-specific BMI for the exposure or sex-specific outcomes.”

(2) I believe that the authors' claim, "To our knowledge, no sex-specific PheWAS has investigated the effects of BMI on health outcomes," is not well supported. They have not cited a relevant paper that conducted both overall and sex-stratified PheWAS using UK Biobank data with a detailed analysis. Given the prior study linked above, I am uncertain about the additional contributions of the present research.

Thank you for your valuable comments, please accept our apologies for this oversight. As explained above, we have checked very carefully. There are three previous PheWAS for BMI, Hyppönen et al., 2019, Millard et al., 2015 and Millard et al. 2019. Hyppönen et al., 2019 and Millard et al., 2015 are not sex-specific. Millard et al. 2019 used sex-combined instruments, but some sex-specific outcomes, when the questions were asked sex-specifically, such as age at puberty asked as “age when periods started (menarche)” in women and “relative age of first facial hair” and “relative age voice broke” in men. When they found a factor significantly associated with BMI, they sometimes analyze it further including sex-specific analysis, but they did not do the analysis systematically for men and women with sex-specific BMI and sex-specific outcomes. We have amended the introduction to clarify this point.

“To our knowledge, no sex-specific PheWAS has investigated the effects of BMI on health outcomes (Hyppönen et al., 2019; Millard et al., 2015; Millard et al. 2009). To address this gap, we conducted a sex-specific PheWAS, using the largest available sex-specific GWAS of BMI, to explore the impact of sex-specific BMI on sex-specific health-related attributes”

**Recommendations for the authors:**

**Reviewer #1 (Recommendations for the authors):**
Presentation, accuracy, and referencing:(1) The quality of the English language needs to be checked, including that all sentences carry all components required (including verbs).

We thank the reviewer for this suggestion. The manuscript has undergone language editing by a native English-speaker, with particular attention to grammatical completeness (including verb consistency and sentence structure). We have also clarified ambiguities and inconsistencies in terms pointed out by the native English speakers. All revisions have been implemented in the updated manuscript.

(2) The accuracy of statements needs to be checked. For example, in lines 82-83 it is not true that 2015/2019 was 'before the advent of large-scale GWAs studies". In the context of the above in lines 83-85, how can reference be made to a study published in 2020 calling that 'previous' MR studies and how a trial published in 2016 is 'recent'? Please revise, and please also check the manuscript for any other issues with accuracy of this kind.

We thank the reviewer for this suggestion. We have checked the manuscript and revised these sentences to be clearer, by making the following change.

“Previous phenome-wide association studies using MR (MR-PheWASs) have identified impacts of sex-combined BMI on endocrine disorders, circulatory diseases, inflammatory and dermatological conditions, some biomarkers and feelings of nervousness (Hyppönen et al., 2019; Millard et al., 2015; Millard et al. 2019), but did not systematically use sex-specific BMI for the exposure or sex-specific outcomes. Previous MR studies and trials of incretins have expanded our knowledge about a broad range of effects of BMI (Larsson et al., 2020; Marso et al., 2016).”

(3) The adequacy of referencing will need to be checked, e.g. line 136 "as recommended by UK biobank" is vague and needs to be referenced.

We thank the reviewer for this suggestion. We have added citations.

“We categorized attributes as age at recruitment, physical measures, lifestyle and environmental, medical conditions, operations, physiological factors, cognitive function, health and medical history, sex-specific factors, blood assays and urine assays, based on the UK Biobank categories (https://biobank.ndph.ox.ac.uk/ukb/cats.cgi).”

(4) The accurate use of terminology needs to be checked. For example, BMI is a measure of adiposity, while high BMI (typically >30) is used to index obesity.

We thank you for your comments. We have changed the descriptions into “overweight/obesity” throughout.

(5) Figure 1, Please check that complete information is given for 'selection criteria' and that the rationale for all information included is clear. For example, it is currently unclear what is the distinction between the bottom two sections which both present a number of features included in the analyses? Also, the Box detailing exclusion of 3585 variables does not give the criteria for these exclusions. Please add.

Thank you for your comments. We have represented and revised Figure 1. Specifically, we have revised the bottom two sections to give each reason for exclusion and the number excluded for that reason. The updated “Excluded: 3,572 phenotypes, for the reason listed below:” box now contains bullet-points giving each reason for exclusion in the box (e.g. age of certain diseases/disorders onset: 26, alcohol: 56).

(6) Figure 4, does not look to be of typical publication quality.

We thank you for your comments. We have used different colors to make it smaller and more readable. Please see Table 1.

Analyses:(1) As it stands, it is very difficult for a reader to confirm the conclusion that similar findings are obtained both when using instruments from the UKB and GIANT based on data presented (Stable 1 and 2). I suggested two things.a) Organise stable 1 and 2 by significance and category, with separation by highlighting for those which are significant under correction. I would consider merging these two tables, such that it would be easy for the reader to make the comparisons side by side. Consider presenting separate tables for the analyses for women and men.

We thank you for your comments. We have followed your helpful advice and merged S Table 1 and S Table 2 into S Table 1. Furthermore, we have also merged S Table 5 to S Table 1.

b) In Stable 3, please add information from related comparisons using the GIANT instruments. To support the authors' claim that associations are similar, but only the precision of estimation differed, you could consider adding information for numbers of associations for those that are directionally consistent and which have an association at least under nominal significance'. For associations where this does not hold, I would refrain from making a claim that the results are not affected by the choice of instrument (or biases relating to the analysis conducted).

We thank you for your comments. Among 42 significant sex-specific associations identified in both the UK Biobank and the sex-specific GIANT consortium for men, all showed consistent directions of effect. Similarly, for women, all of the 45 significant associations exhibited consistent directions for UK Biobank compared with GIANT instruments.

In the sex-specific UK Biobank, there are 203 significant associations in men, and 232 significant associations in women. We have added: in the sex-specific GIANT, there are 46 significant associations in men, and 84 significant associations in women. In the sex-combined GIANT, there are 246 significant associations in men, and 276 significant associations in women. We have provided all this information in S Table 2.

We added the following descriptions at the end of the results section:

“Of the 42 significant sex-specific associations identified in both the UK Biobank and the sex-specific GIANT consortium for men, all were directionally consistent. Similarly, for women, all 45 such significant associations were directionally consistent.

We amended the following descriptions in the first paragraph of the results section:

“Initial analysis using sex-specific BMI from the GIANT yielded similar estimates as when using sex-specific BMI from the UK Biobank but had fewer SNPs resulting in wider confidence intervals (S Table 1) and fewer significant associations (S Table 2). Analysis using sex-combined GIANT yielded more significant associations but lacks granularity, so we presented the results obtained using sex-specific BMI from the UK Biobank.”

In the methods, we changed:

“We obtained differences by sex using a z-test (Paternoster et al., 1998), which as recommended was on a linear scale for dichotomous outcomes (Knol et al., 2007; Rothman, 2008), then we identified which ones remained after allowing for false discovery.”

We have made the following changes to the results section:

“We found significant differences by sex in the associations of BMI with 105 health-related attributes (p-value<0.05); 46 phenotypes remained after allowing for false discovery (Table 1). Of these 46 differences most (35) were in magnitude but not direction, such as for SHBG, ischemic heart disease, heart attack, and facial aging, while 11 were directionally different.

Notably, BMI was more strongly positively associated with myocardial infarction, major coronary heart disease events, ischemic heart disease, heart attack, and facial aging in men than in women. BMI was more strongly positively associated with diastolic blood pressure, and hypothyroidism/myxoedema in women than men. BMI was more strongly inversely associated with LDL-c, hay fever and allergic rhinitis in men than women. BMI was more strongly inversely associated with SHBG in women than men.

BMI was inversely associated with ApoB, iron deficiency anemia, hernia, and total testosterone in men, while positively associated with these traits in women (Table 1). BMI was inversely associated with sensitivity/hurt feelings, and ever seeking medical advice for nerves, anxiety, tension, or depression in men. However, BMI was positively associated with sensitivity/hurt feelings and ever seeking medical advice for these same issues in women. BMI was positively associated with muscle or soft tissue injuries and haemorrhage from respiratory passages in men, whilst inversely associated with these traits in women.”

(2) It is not clear what statistical criteria were used to determine sex differences, and the strategy/presentation should be clarified. In lines 229-231, it is implied that the 'significance' in one gender, but not in the other is used to indicate a difference. However, 'comparison of p-values' is not a valid statistical approach, and a more formal test (accounting for multiple testing would be warranted). It may be that a systematic approach has been implemented, but please check that it is adequately and accurately described to the reader.

Please accept our apologies for being unclear. Multiple comparisons are for independent phenotypes however, here, some phenotypes cannot be independent, therefore, using multiple comparisons in men and women separately is quite strict. We added multiple comparisons for the assessment of sex-differences, which is now given in Table 1. Initially, there were 105 significant associations (p value for sex-difference<0.05) (Table 1), and 46 associations remained after FDR correction (Table 1).

Furthermore, we have made additional minor changes to clarify the wording.

Knol, M. J., van der Tweel, I., Grobbee, D. E., Numans, M. F., & Geerlings, M. I. (2007). Estimating interaction on an additive scale between continuous determinants in a logistic regression model. Int J Epidemiol, 36(5), 1111-1118.

Nikiforov, S. V., & Mamaev, V. B. (1998). The development of sex differences in cardiovascular disease mortality: a historical perspective. Am J Public Health, 88(9), 1348-1353. https://doi.org/10.2105/ajph.88.9.1348

Paternoster, R., Brame, R., Mazerolle, P., & Piquero, A. (1998). Using the correct statistical test for the equality of regression coefficients. Criminology, 36(4), 859-866.

Rothman, K. (2008). Greenland S, Lash TL (ed.). Modern Epidemiology. In: Philadelphia: Lippincott Wolliams & Wilkins.